# Emergency department returns and early follow-up visits after heart failure hospitalization: Cohort study examining the role of race

**Rachel E. Solnick**[1,2,3]*, **Ganga Vijayasiri**[4], **Yiting Li**[4], **Keith E. Kocher**[1,3,5], **Grace Jenq**[4,6], **David Bozaan**[4,7,8]

**1** Department of Emergency Medicine, School of Medicine, University of Michigan, Ann Arbor, MI, United States of America, **2** Now at Department of Emergency Medicine, Icahn School of Medicine at Mount Sinai, New York, NY, United States of America, **3** Institute for Healthcare Policy and Innovation, University of Michigan, Ann Arbor, MI, United States of America, **4** Integrated Michigan Patient-Centered Alliance in Care Transitions (I-MPACT), Michigan Medicine, Ann Arbor, MI, United States of America, **5** Department of Learning Health Sciences, School of Medicine, University of Michigan, Ann Arbor, MI, United States of America, **6** Division of Geriatric and Palliative Medicine, Department of Internal Medicine, School of Medicine, University of Michigan, Ann Arbor, MI, United States of America, **7** Division of Hospital Medicine, Department of Internal Medicine, Michigan Medicine, Ann Arbor, Michigan, United States of America, **8** Veterans Affairs Ann Arbor Healthcare System, Ann Arbor, Michigan, United States of America

* Rachel.Solnick@mountsinai.org

**Data Availability Statement:** Data cannot be shared publicly because of contractual agreements between participating institutions and the I-MPACT (Integrated Michigan Patient-centered Alliance in

## Abstract

Health disparities in heart failure (HF) show that Black patients face greater ED utilization and worse clinical outcomes. Transitional care post-HF hospitalization, such as 7-day early follow-up visits, may prevent ED returns. We examine whether early follow-up is associated with lower ED returns visits within 30 days and whether Black race is associated with receiving early follow-up after HF hospitalization. This was a retrospective cohort analysis of all Black and White adult patients at 13 hospitals in Michigan hospitalized for HF from October 1, 2017, to September 30, 2020. Adjusted risk ratios (aRR) were estimated from multivariable logistic regressions. The analytic sample comprised 6,493 patients (mean age = 71 years (SD 15), 50% female, 37% Black, 9% Medicaid). Ten percent had an ED return within 30 days and almost half (43%) of patients had 7-day early follow-up. Patients with early follow-up had lower risk of ED returns (aRR 0.85 [95%CI, 0.71–0.98]). Regarding rates of early follow-up, there was no overall adjusted association with Black race, but the following variables were related to lower follow-up: Medicaid insurance (aRR 0.90 [95%CI, 0.80–1.00]), dialysis (aRR 0.86 [95%CI, 0.77–0.96]), depression (aRR 0.92 [95%CI, 0.86–0.98]), and discharged with opioids (aRR 0.94 [95%CI, 0.88–1.00]). When considering a hospital-level interaction, three of the 13 sites with the lowest percentage of Black patients had lower rates of early follow-up in Black patients (ranging from 15% to 55% reduced likelihood). Early follow-up visits were associated with a lower likelihood of ED returns for HF patients. Despite this potentially protective association, certain patient factors were associated with being less likely to receive scheduled follow-up visits. Hospitals with lower percentages of Black patients had lower rates of early follow-up for Black patients. Together, these may represent

Care Transitions) CQI (Collaborative Quality Initiative). Data are available from the I-MPACT Program Manager (contact via i-mpactcc@med. umich.edu) for researchers who meet the criteria for access to confidential data.

**Funding:** The Blue Cross Blue Shield of Michigan and the Blue Care Network as part of the BCBSM Value Partnerships program helped support the data collection and staff used to develop this work. The funders had no role in study design, analysis, decision to publish, or preparation of the manuscript.

**Competing interests:** The authors have declared that no competing interests exist.

missed opportunities to intervene in high-risk groups to prevent ED returns in patients with HF.

## Introduction

Black patients face the worst heart disease survival rates of all racial groups [1], and Black patients, in particular, experience higher age-adjusted HF death rates compared to White patients [2]. Studies have found racial disparities in the clinical management of HF, with Black patients having reduced access to HF care: Black HF patients compared to Whites were less likely to see a cardiologist while in the intensive care unit [3], less likely to be hospitalized from the ED for HF [4], and less likely to receive evidence-based advanced HF treatment, including an automatic implantable cardioverter-defibrillator (AICD) implantation [5–7] or cardiac resynchronization therapy with defibrillation (CRT-D) [8, 9]. These racial disparities in access to HF care may be partly explained by implicit bias, as has been examined in vignette studies: in one interview study, race influenced healthcare professionals' rationale for why a Black patient was not considered a candidate for a heart transplant [10], and in a photo-based study, Black women were less likely to be referred for cardiac catheterization [11].

In addition to racial disparities in HF mortality, Black patients also have greater ED utilization and face higher readmission rates [12–15]. Higher healthcare utilization among these groups may represent an opportunity for additional services such as transitional care—specifically early follow-up visits—to reduce health disparities in groups that either through comorbidities are medically vulnerable, or through structural racism and economic forces, are medically underserved. Defined as services a patient receives when changing healthcare settings and including telephone follow-ups or early follow-up visits, transitional care, and specifically 7-day follow-up visits, have been associated with reduced hospital readmissions following HF hospitalization [16–18]. Limited evidence suggests certain transitional care measures such as telephone follow-up and structured home visits are associated with reduced ED returns following HF hospitalization [19]. ED return after HF discharge is a potentially avoidable event that frequently results in hospitalization [20], is a marker of clinical exacerbation in HF symptoms, and creates fragmentation in patient care. Thus, early follow-up visits after hospitalization may represent a measurable, modifiable, process-based quality measure that could be harnessed to address racial disparities in HF outcomes.

Despite their potential to decrease unplanned healthcare utilization, less is known about whether early follow-up visits are associated with reduced ED returns and whether race is associated with early follow-up visits. Thus, we first examine whether early follow-up is associated with lower 30-day ED returns. Secondly, we evaluate whether Black race is associated with receiving a scheduled 7-day follow-up after discharge from a HF hospitalization. Because transitional care practices may vary by hospital, we further examine whether Black race is associated with early follow-up rates at the hospital-level. Since prior literature demonstrates the greatest disparities between White and Black races, and our aim in examining racial disparities is to offer transitional care interventions, we restrict our analysis sample to these groups.

We hypothesized that early follow-up is associated with lower ED returns and that Black patients are less likely to have an early follow-up scheduled overall, but that there would be lower rates of follow-up at hospitals with less Black patients. We conduct this study in a quality improvement collaborative of 13 hospitals across Michigan which provide data on a set of targeted conditions, including HF. Hospital participation was voluntary. This unique multi-payer database allows for the examination of care across a variety of hospital settings with all-payer patient data for HF.

## Methods

We performed a retrospective cohort analysis of all Black and White adult patients at 13 hospitals in Michigan hospitalized for HF from October 1, 2017, to September 30, 2020. Details of the participating hospitals are as follows: all non-profit and general acute care hospitals, 6 church ownership, 7 private ownership, 12 teaching hospitals. Number of hospital beds ranged from 129–637 beds, 0 critical access designated. Race was captured via self-report using the categories White or Caucasian, Black or African American, Asian, Native Hawaiian or Other Pacific Islander, American Indian or Alaska Native, and some other race. Of the initial sample of 6,821 individuals, those with missing data were removed from the analysis (n = 328). Missing data ranged from 0% to 1.89%; missingness from key variables was as follows: ED return visits 0%, follow-up visits 0.03%, and race 1.89%. Eligibility criteria for chart abstraction were as follows: not pregnant, did not die during hospitalization, not under the age of 18, and did not leave the hospital against medical advice.

Given the inclusion of data captured during the COVID-19 pandemic, a sensitivity analysis was conducted excluding data from the most affected period. The main regression models were re-fitted using a sample that excluded patients discharged from April 1, 2020, through the end of sample data on September 30, 2020.

Charts were manually abstracted by trained chart abstractors as part of a quality improvement initiative funded by Blue Cross Blue Shield of Michigan (BCBSM) and Blue Care Network. Abstractors were initially trained by quality improvement nursing staff from the BCBSM central coordinating center following detailed data element guides. For each participating hospital, yearly audits of three charts and abstracted elements were conducted by the centralized quality improvement nurses to ensure quality control. The audits are a part of the performance improvement incentive score. If data abstractors abstract elements incorrectly, points are deducted and retraining occurs with correction of the elements, and the chart is re-audited. Although funded by BCBSM, the charts analyzed were from all patients hospitalized for HF, regardless of the health insurance carrier.

### Statistical analysis

The main outcome measures were ED returns, defined as a visit to the ED within 30 days of discharge and early follow-up, defined as a scheduled appointment within 7 days of hospital discharge. Early follow-up visits were incentivized by BCBSM through pay-for-performance and were counted if scheduled. The association between early follow-up and 30-day ED returns was assessed using a multivariable logistic regression model. A stepwise procedure was used to select covariates into the multivariable model from a list of 21 demographic and clinical variables (S1 File) selected by a theory-driven approach of variables related to healthcare utilization for HF [16–18, 21], and described in Table 1. Stepwise procedure selected covariates using a p-value threshold of 0.1 and selected the variable of interest (early follow-up); four LACE [22] risk scoring items (length of stay, acuity of admission, Charlson comorbidity index, and prior emergency department visits), and receipt of additional transitional care activities (coded 0 and 1) regardless of their statistical significance. The Hosmer-Lemeshow goodness of fit test indicated good fit ($P = 0.276$) for this model (Model 1).

A multivariable logistic model was also estimated to assess whether Black race was associated with a lower early follow-up rate (Model 2). Again, stepwise logistic regression was used to select covariates into the multivariable model from demographic and clinical variables (described in Table 1), with stepwise procedure retaining the variable of interest (race), Medicaid, four LACE risk scoring items, prior hospitalizations regardless of their statistical significance, and other variables using a p-value threshold of 0.1. While Black race and Medicaid

**Table 1. Characteristics of Black and White patients hospitalized for heart failure at 13 hospitals in Michigan from 2017 to 2020.**

| | Overall | Early Follow-up | No Early Follow-up | p-value |
|---|---|---|---|---|
| | (N = 6,493) | (N = 2,798) | (N = 3,695) | |
| **Outcome** | | | | |
| ED return visits within 30 days | 10% | 9% | 11% | 0.039 |
| **Sociodemographics** | | | | |
| Black | 37% | 31% | 42% | <0.001 |
| Age (SD) | 71 (15) | 72 (14) | 70 (15) | <0.001 |
| Female | 50% | 49% | 51% | 0.204 |
| Married | 41% | 46% | 37% | <0.001 |
| Medicaid | 9% | 7% | 10% | <0.001 |
| Neighborhood Income ($1000) (SD) [a] | 52.0(20.7) | 53.5(19.6) | 51.7(21.5) | 0.11 |
| **Clinical Characteristics: Patient** | | | | |
| Charlson Comorbidity Index (SD) [b] | 4.2(2.2) | 4.3(2.2) | 4.1(2.1) | <0.001 |
| Diabetes uncontrolled | 9% | 10% | 8% | 0.028 |
| Discharged with > = 10 medications | 67% | 70% | 64% | <0.001 |
| Discharged with Opioids | 24% | 24% | 25% | 0.151 |
| Discharged on Antiplatelets | 63% | 61% | 64% | 0.006 |
| Depression | 27% | 28% | 27% | 0.48 |
| Require Dialysis | 9% | 7% | 10% | <0.001 |
| **Clinical Characteristics: Hospitalization** | | | | |
| ED visits in prior 180 days | 32% | 33% | 32% | 0.486 |
| Admissions in prior 180 days | 49% | 48% | 50% | 0.057 |
| Treated in Intensive Care Unit | 8% | 8% | 8% | 0.877 |
| PCP identified in Discharge Summary [c] | 86% | 90% | 83% | <0.001 |
| DS/AVS medication discrepancy [c] | 53% | 46% | 58% | <0.001 |
| Length of Stay (SD) [c] | 4.7(3.6) | 4.8(3.3) | 4.8(3.8) | 0.786 |
| Received other transitional care/s [d] | 70% | 80% | 63% | <0.001 |
| Admitted from ED [c] | 90% | 88% | 91% | <0.001 |

P-values for categorical patient measures are based on chi-square tests, while p-values for continuous measures are based on T-tests

[a]Neighborhood income is zip code median household income measured in units of $1K

[b]Charlson Comorbidity Index (1–12)

[c]Abbreviations: ED, Emergency Department; PCP, Primary Care Provider; AVS, After Visit Summary; DS, Discharge Summary (d) Other quality improvement transitional care intervention(s) intervention(s) determined at the cluster level excluding 7 day follow up, activities such as care management, standardized discharge process, consults/ referrals, patient education, or medication reconciliation.

were moderately correlated (0.235), collinearity diagnostics indicated lack of collinearity between the two variables (Black: VIF = 1.15, Tolerance = 0.873; Medicaid: VIF = 1.09, Tolerance = 0.920).

To test the hypothesis that the association between Black race and early follow-up may vary across hospital sites, we re-estimated Model 2 adding an interaction term between Black race and site indicators (Model 3). After fitting the model with interaction terms, predicted probability of early follow-up for Black patients were calculated taking the interaction effect into account. While the Hosmer-Lemeshow goodness of fit test indicated good fit for both models (Model 2: $P = 0.483$; Model 3: $P = 0.829$), a model comparison confirmed that the larger model (Model 3) had better fit.

Using the logistical regression models cited above, we calculated the Adjusted Risk Ratios (aRRs), a measure of relative risk, for ease of interpretation of the logit model. The ARR is a

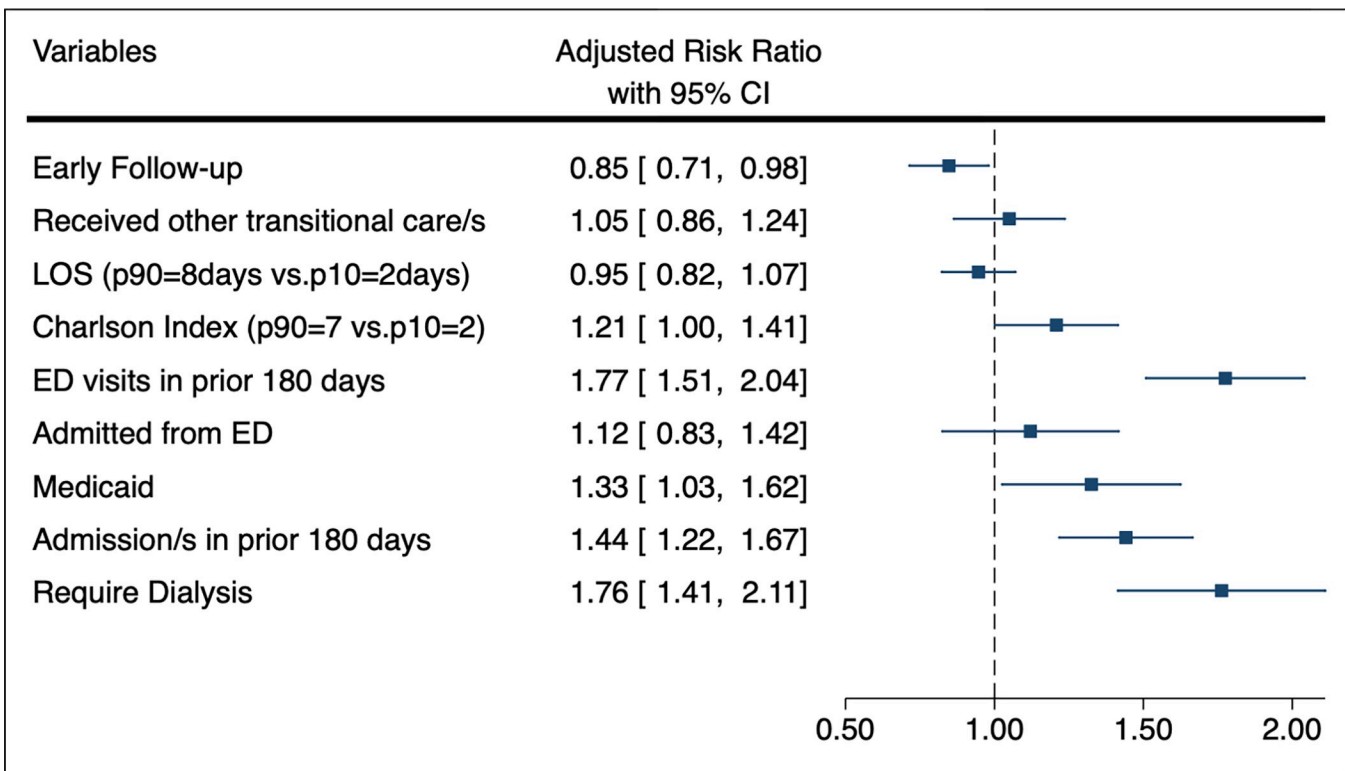

**Fig 1. Adjusted associations between early follow-up and ED Returns in Black and White patients hospitalized for heart failure at 13 hospitals in Michigan from 2017 to 2020.** Adjusted Risk Ratios (aRRs) calculated from multivariable logistic regression model between Early Follow-up and ED returns in addition to all other included covariates. ED Returns is defined as ED visit within 30-days of HF hospital discharge; early follow-up is defined as scheduled follow-up visit within 7 days.

postestimation calculation which estimates predicted probabilities based on the logit model. The fit model is used to estimate the ratio of the adjusted predicted risk among the exposed to the adjusted predicted risk among those unexposed. Thus, the ARR is the ratio of the mean of theses predicted probabilities averaged over the entire dataset. We conducted this analysis using the adjrr command in Stata [23].

Fig 1 displays Model 1 for the association between factors and ED returns. Fig 2 –Panel A reports aRRs for Model 2 for the association between factors and early follow-up. Fig 2 –Panel B reports aRRs from Model 3 for the association between Black race and early follow-up at each of the 13 hospital sites. For descriptive purposes, Fig 2 –Panel B also provides percentage of Black patients in each of the 13 sites.

Analyses were performed using STATA v16 (StataCorp). The University of Michigan Institutional Review Board (IRB) waived review of this study. The data were fully anonymized before being accessed and the IRB waived the requirement for informed consent.

## Results

The analytic sample comprised 6,493 patients with a mean age of 71 years (SD 15), 50% female, 37% Black, and 9% Medicaid (Table 1 and stratified by race in S1 Table). Most HF hospitalized patients (90%) were admitted from the ED, 8% were treated in the intensive care unit during the stay, and the average length of stay was 4.7 days (SD 3.6). Ten percent of patients (n = 666) had an ED return visit within 30 days. Almost half (43%, n = 2,798) of patients had an early

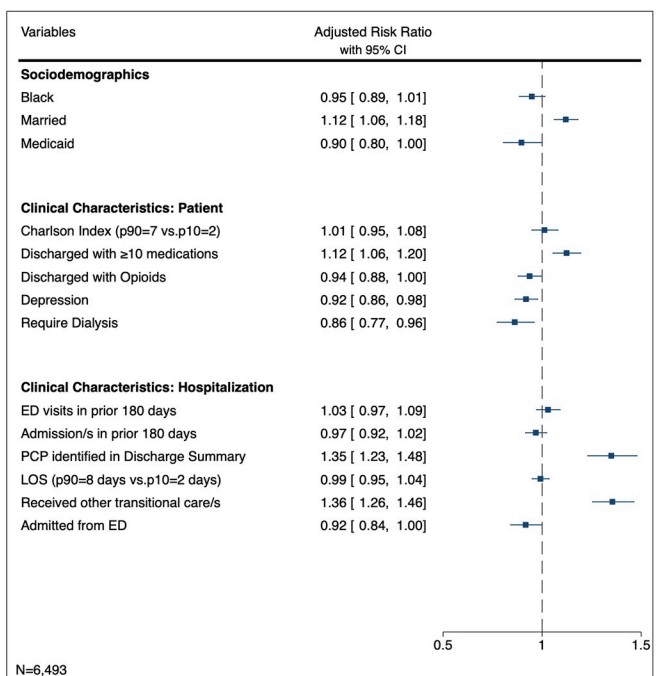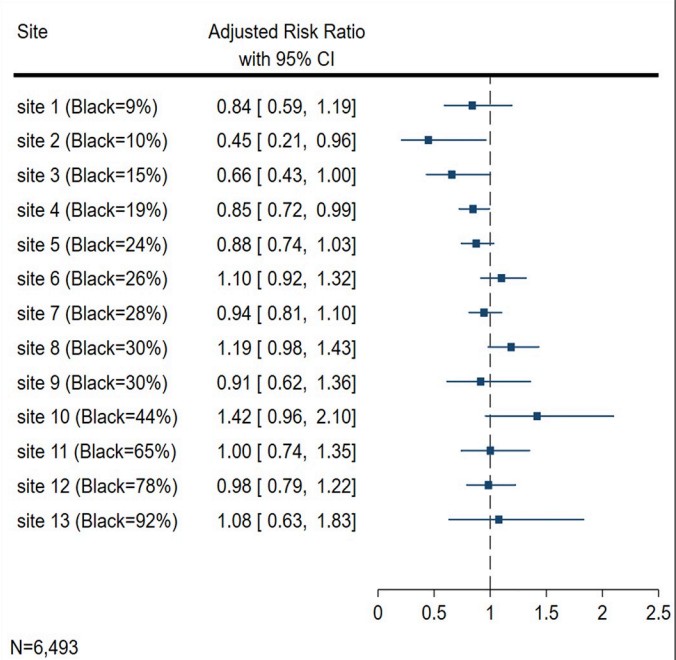

**Fig 2. Adjusted association between race and early follow-up by aggregate and by hospital site in Black and White patients hospitalized for heart failure at 13 hospitals in Michigan from 2017 to 2020.** Notes: Adjusted Risk Ratios (ARRs) calculated from multivariable logistic regression model between early follow-up (defined as scheduled 7-day follow-up) and all other included covariates. Panel A shows the model with the Black race variable and no site interaction term. Panel B shows the model that included all Panel A variables with an additional race x site interaction term. We present the predicted adjusted association between Black race and 7-day follow-up by each hospital site ordered by percentage of Black patients hospitalized for HF in each site. Results listed in tabular form in S3 Table.

follow-up visit scheduled within 7 days. Patients with no early follow-up visits compared to those who had early follow-up were more likely to have an ED return within 30 days (11% versus 9%, $P = 0.039$) and were more likely to be Black (42% versus 31%, $P = <0.001$).

## Emergency department returns

The relationship between early follow-up visits and ED returns is displayed graphically with covariates in Fig 1, and as unadjusted results in the S2 Table. Adjusted analysis found that patients with early follow-up visits had a 15% lower risk of ED returns (aRR 0.85 [95%CI, 0.71–0.98]) compared with patients who did not have early follow-up visits. The following factors showed an increased risk of ED returns: Medicaid insurance (aRR 1.33 [95%CI, 1.03–1.62]), prior admission in last 180 days (aRR 1.44 [95%CI, 1.22–1.67]); and dialysis (aRR 1.76 [95%CI, 1.41–2.11]).

## Early follow-up across all hospitals

The relationship between race and early follow-up visits is displayed graphically with covariates in Fig 2 Panel A (tabular form in S3 Table) and as unadjusted results in S4 Table. The unadjusted analysis found Black patients had a reduced likelihood of early follow-up compared to White patients (RR 0.76 [95%CI, 0.71–0.80]). The adjusted analysis found Black patients had no significant association with early follow-up (aRR 0.95 [95%CI, 0.89–1.01]) compared to White patients. Several patient clinical factors were associated with lower rates of early follow-up: dialysis (aRR 0.86 [95%CI, 0.77–0.96]), Medicaid insurance (aRR 0.90 [95%CI, 0.80–

1.00]), depression (aRR 0.92 [95%CI, 0.86–0.98]), and being discharged with opioids (aRR 0.94 [95%CI, 0.88–1.00]).

### Early follow-up by hospital-level

The relationship between race and early follow-up visits by hospital-level is displayed graphically with covariates in Fig 2 Panel B (tabular form in S3 Table). The adjusted analyses found that three hospitals that had a lower proportion of HF patients of Black race also had lower rates of early follow-up for Black patients, ranging from 15%-55% reduced likelihood of early follow-up in Black patients. The hospitals with the lowest two follow-up rates for Black patients were at a site with 10% Black patients (aRR 0.45 [95%CI, 0.21–0.96]), and a site with 15% Black patients (aRR 0.66 [95%CI, 0.43–1.00]).

Sensitivity analysis on both main outcomes excluding COVID-19 affected quarters (discharges from April 1, 2020, through the end of sample data on September 30, 2020) showed findings consistent with the main analysis models. There were slightly stronger associations between early follow-up and lower risk of ED return (aRR 0.78 [95%CI 0.65–0.93]) compared to the full 2020 dataset. As in the full 2020 model, we did not observe any statistically significant overall association between Black race and early follow-up when we excluded COVID-19 affected quarters.

## Discussion

Having an early follow-up visit after HF hospitalization was associated with a lower likelihood of an ED return. Despite this potentially protective association, certain patient factors reflective of clinical vulnerability (dialysis, depression, discharge on opioid prescriptions) were associated with being less likely to receive early follow-up. At select hospital sites with lower proportions of Black HF patients, Black race was also associated with a lower likelihood of early follow-up.

There are a few potential explanations and interpretations for these findings. Early follow-up visits after HF hospitalization could avert an ED visit due to the medical interventions and coordination of care that takes place during the follow-up visit. While it may be expected that patients with more comorbidities or difficulty with healthcare access—such as patients on dialysis with Medicaid insurance—should be targeted for earlier follow-up since they are at higher risk for readmission and unscheduled healthcare needs, these higher-risk patients were less likely to receive early follow-up visits in this study. Specifically, Medicaid insurance and receipt of dialysis were factors associated with a lower likelihood of early follow-up but also a higher likelihood to have an ED return. One explanation for this observation may be that these patients fared poorly due to the lack of follow-up and thus needed more urgent care in the ED. Our theory—that transitional care can avert ED returns—is supported by previous randomized trials based in Canada, which found that HF patients randomized to transitional care after discharge had a reduction in rate of ED returns [19]: in one study in 2000 by Harrison et al. of 200 patients, nurse-led transitional care of structured home visits and telephone follow-up reduced ED visits; in the second study in 1998 by Tsuyuki et al. of 276 patients, transitional care including patient education and telephone follow-up reduced cardiovascular-related ED visits. These findings speak to the inherent value of prompt medical evaluation after HF hospitalization.

An alternate explanation is that these patients are less likely to schedule outpatient appointments due to difficulty navigating the healthcare system [24], and thus are more likely to access healthcare via unscheduled methods such as the ED. Of note, Medicaid insurance accounts for a disproportionately higher number of ED visits than other insurance groups [25], a finding

also seen in our population—the ED visit rate for Medicaid patients was 14.5% (n = 82/567) versus 9.8% for non-Medicaid patients (n = 584/5926) ($\chi 2$ for difference p<0.001). Importantly, the variable of follow-up visit was measured if the case was scheduled, not completed, and so transportation factors, a known barrier to accessing healthcare, are not relevant. Thus, patterns of healthcare usage may be a confounder in interpreting links between factors related to lower early follow-up and higher ED returns.

Both explanations suggest the need for higher-touch outreach in scheduling early follow-up and/or increased coordination of care before discharge for patients identified as at greater risk of ED returns. Previous studies have found that higher-intensity transitional care, such as telephone follow-up with home visits, had a greater impact on reducing readmission compared to lower-intensity measures, such as telephone follow-up alone [19]. Additionally, a meta-analysis found nurse home visits were more effective than disease-management clinics at reducing readmissions [26]. Multidisciplinary teams including allied health professionals, navigators, and community health workers can play a role in facilitating early follow-ups [27]. Future studies should examine the impact of such higher-intensity interventions for groups less likely to schedule follow-ups and/or more likely to have ED returns.

Our findings extend previous findings linking transitional care with lower ED returns. A meta-analysis of transitional care interventions that included 5 randomized control trials found a reduced risk of ED returns (relative risk 0.71 [95%CI, 0.51–0.98]), which was slightly greater in magnitude than in our study [19]. The studies in this meta-analysis with the greatest reduction in ED visits utilized structured home visits and telephone follow-up, as opposed to our study, which only examined clinic-based follow-up visits. In a stepped-wedged cluster randomized trial of 2,494 patients in Canada which examined transitional care interventions including a 7-day follow-up visit, the exploratory outcome of 30-day ED returns was reduced in the transitional care arm (HR 0.65 [95%CI 0.45–0.95]) compared to standard care [28].

While we examined ED returns within 30 days as a measure of care utilization, more evidence exists on the impact of transitional care on hospital readmission among HF patients. Since the majority of HF hospitalizations originated from ED admissions in this study (90%), and nationally, most (83%) ED visits for HF are admitted, comparing our study to the readmission literature is instructive. In a meta-analysis of transitional care studies, visiting a disease management clinic after HF hospitalization reduced all-cause readmission of varying periods post-discharge (incident rate ratio 0.80, 95% CI 0.66–0.97) [26]. At a hospital level, a study of 225 hospitals found a significant decrease in 30-day readmissions for hospitals with the highest early follow-up rates compared to the lowest (adjusted hazard ratio 0.91 95%CI 0.83–1.00) [16]. A case-control study of 11,985 HF patients within an integrated healthcare system in California also found a reduced 30-day readmission for patients with 7-day follow-up compared to those who received later appointment dates (aOR 0.81, 95%CI 0.7–0.94) [17]. Lastly, a study of 13,577 HF patients in Taiwan demonstrated a reduced hazard ratio of 30-day readmission for patients receiving 7-day physician follow-up (HR 0.54, 95%CI 0.48–0.6) [18].

Additionally, we find that although Black patients have worse HF outcomes, hospitals with lower proportions of Black HF patients were less likely to schedule an early follow-up for Black patients. That Black patients were less likely to have follow-ups after HF hospitalization is consistent with the findings from a previous Medicare claims analysis [29]. However at an individual hospital level, our finding is surprising given Medicare data which shows Black patients at non-minority serving hospitals had lower readmission rates (23%) than Blacks at minority-serving hospitals (26%), seeming to imply better outcomes for Black patients at hospitals with lower proportions of Black patients [30]. One might expect the non-minority-serving hospital would then increase rates of scheduling for early follow-up in Black patients to decrease racial disparities in care, however, this is not what we observed. These disparities in scheduling

follow-up care existed despite a hospital network wide quality improvement program that specifically financially incentivized these visits. As an observational study, the explanations for this finding are uncertain. Possible explanations of reduced access to HF care by Black patients include: implicit bias—consistent with findings of racial disparities in the management and evaluation of Black HF patients [3, 5–9, 12], and residual confounding from social variables such as educational attainment and poverty [31].

Together, these findings may represent missed opportunities to intervene in high-risk groups to prevent ED returns. Study limitations include: experiences in Michigan may not generalize to other states, missing information on whether scheduled follow-up visits actually occurred, unknown ED return or early follow-up data if a patient went to an ED, hospital, or clinic outside of the 13 hospitals included in the study, and unmeasured confounding from neighborhood effects and other social determinants of health. Furthermore, the reason for the ED return was not delineated and could be related to other comorbidities as opposed to HF. Study strengths include: detailed patient-level and visit-level variables to help understand contextual factors of healthcare utilization and high-quality chart abstracted all-payer data across a wide variety of hospitals situated in geographic areas with diverse demographics.

As we recognize the human costs of our fragmented healthcare system, our findings underscore the importance of identifying disparities in HF follow-up care and the necessity to improve the equity of care for those in the highest need. Investigations into transitional care measures are increasingly important given the potential negative influence of national policies that may not adequately consider sociodemographic and community factors in their scoring systems [32, 33]. One such example is the Hospital Readmissions Reduction Program (HRRP), a Centers for Medicare & Medicaid Services (CMS) program that does not incorporate social determinants [34, 35]—factors which may influence healthcare utilization and outcomes—but penalizes hospitals for higher than expected rates of readmission. The policy implications of our findings highlight the need to measure, monitor, and mitigate the potential inequitable effects of national quality measures, especially those that could exacerbate financial challenges for already struggling safety-net hospitals. Incorporating modifiable, process-based measures, such as equity in the offering of transitional care services, may represent a future model for federal quality measure consideration.

## Supporting information

**S1 File. Explanatory variable operational definitions.**
(DOCX)

**S1 Table. Patient characteristics stratified by race.** P-values for categorical patient measures are based on chi-square test, while p-values for continuous measures are based on linear regression. [a] Neighborhood income is zip code median household income measured in units of $1K. [b] Charlson Comorbidity Index (1–12) [c] Abbreviations: ED, Emergency Department; PCP, Primary Care Provider; AVS, After Visit Summary; DS, Discharge Summary. [d] Other quality improvement transitional care intervention(s): intervention(s) determined at the cluster level excluding 7-day follow-up, activities such as care management, standardized discharge process, consults/referrals, patient education, or medication reconciliation.
(DOCX)

**S2 Table. Unadjusted associations between early follow-up and emergency department.** (a) Comparison is for 10th and 90th percentile values. (b) Abbreviations: ED, Emergency Department; PCP, Primary Care Provider; AVS, After Visit Summary; DS, Discharge Summary (c) Other quality improvement transitional care intervention(s): intervention(s) determined at

the cluster level excluding 7-day follow-up, activities such as care management, standardized discharge process, consults/referrals, patient education, or medication reconciliation. (DOCX)

**S3 Table. Adjusted associations between patient factors and early follow-up by aggregate and by hospital site.** Adjusted Risk Ratios (aRRs) from multivariable logistic regression model between early follow-up (scheduled within 7 days post-discharge) and patient/site factors. Panel A shows model with Black race variable and no site interaction term. Panel B shows model that included Panel A variables with an additional race x site interaction term. We present the predicted adjusted association between Black race and 7-day follow-up by each hospital site ordered by percentage of Black patients hospitalized for heart failure in each site. (DOCX)

**S4 Table. Unadjusted associations between patient characteristics and early follow-up.** (a) Comparison is for 10th and 90th percentile values. (b) Abbreviations: ED, Emergency Department; PCP, Primary Care Provider; AVS, After Visit Summary; DS, Discharge Summary (c) Other quality improvement transitional care intervention(s): intervention(s) determined at the cluster level excluding 7-day follow-up, activities such as care management, standardized discharge process, consults/referrals, patient education, or medication reconciliation. (DOCX)

## Author Contributions

**Conceptualization:** Rachel E. Solnick, Ganga Vijayasiri, Yiting Li, Grace Jenq, David Bozaan.

**Data curation:** Yiting Li.

**Formal analysis:** Ganga Vijayasiri, Yiting Li.

**Investigation:** Rachel E. Solnick, Ganga Vijayasiri, Yiting Li, Keith E. Kocher, Grace Jenq, David Bozaan.

**Methodology:** Rachel E. Solnick, Ganga Vijayasiri, Yiting Li.

**Project administration:** Yiting Li, Grace Jenq, David Bozaan.

**Resources:** Grace Jenq, David Bozaan.

**Supervision:** Rachel E. Solnick, Keith E. Kocher, David Bozaan.

**Writing – original draft:** Rachel E. Solnick, Ganga Vijayasiri, Keith E. Kocher.

**Writing – review & editing:** Rachel E. Solnick, Ganga Vijayasiri, Keith E. Kocher, Grace Jenq, David Bozaan.

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
