## [Decision Letter · Decision Letter 0]

11 Mar 2022

PONE-D-21-38345Factors Associated with Early Follow-Up Visits and Lower Emergency Department Returns in Patients Hospitalized for Heart FailurePLOS ONE

Dear Dr. Solnick,

Thank you for submitting your manuscript to PLOS ONE. After careful consideration, we feel that it has merit but does not fully meet PLOS ONE’s publication criteria as it currently stands. Therefore, we invite you to submit a revised version of the manuscript that addresses the points raised during the review process.

 I believe this paper has scientific merit. However, there is not enough details so to assess its methodological rigor and assumptions made when defining the main study concepts. Please consider reviewers' comments when revising the manuscript. 

We look forward to receiving your revised manuscript.

Kind regards,

Alpamys Issanov

Academic Editor

PLOS ONE

Journal Requirements:

Additional Editor Comments (if provided):

Please additionally submit STROBE checklist with your revised manuscript

Reviewers' comments:

Reviewer's Responses to Questions

**Comments to the Author**

1. Is the manuscript technically sound, and do the data support the conclusions?

Reviewer #1: Partly

Reviewer #2: Yes

2. Has the statistical analysis been performed appropriately and rigorously? 

Reviewer #1: I Don't Know

Reviewer #2: Yes

3. Have the authors made all data underlying the findings in their manuscript fully available?

Reviewer #1: No

Reviewer #2: No

4. Is the manuscript presented in an intelligible fashion and written in standard English?

Reviewer #1: Yes

Reviewer #2: Yes

5. Review Comments to the Author

Reviewer #1: Summary: Solnick et al. has submitted a study in which they examine if Black race and other patient factors are associated with receiving a scheduled 7-day follow-up after HF hospitalization discharge and whether early follow- up is associated with lower 30-day ED returns. They conducted a retrospective cohort analysis of 6493 Black and White patients at thirteen Michigan hospitals. They reported that early follow-up visits were associated with a lower likelihood of ED returns for HD patients.

INTRODUCTION

1. It is unclear what the author means by “worse outcomes”. What types of outcomes occur in these patients.

2. Please explain what “transitional care” is.

3. It is unclear if there is any literature on early follow up visits and reduced ED times. If so, please cite existing literature. If not, please note that there is no current literature on the matter.

4. Please clarify why the authors specifically are examining Black race.

5. It is not enough to say “additional patient factors” in the objective sentence. Please clarify specific factors that the authors are interested in.

6. Please note the hypothesis for early follow up and 30-day ED returns

7. Please provide background information on why it is important to study seven follow ups.

8. Please place the study within the context of Michigan and why this study is being conducted in this location.

METHODS

1. It is unclear how the authors categorized Black and White. How were mixed race individuals categorized for this study? Furthermore, please clarify how Hispanic individuals were categorized. Finally, please provide justification as to why only Black and White individuals were included in the study.

2. Please provide justification as to why the thirteen hospitals were chosen. Additionally, please provide details on these hospitals such as location (urban vs rural), type (public vs private), demographics (race).

3. Was race the only variable that was self reported?

4. The author wrote “Early follow-up visits were incentivized by BCBSM through pay-for-performance and were counted if scheduled”. It is unclear if all patients received this incentive or only those that had BCBSM insurance. If the latter, this should be taken into account in the analysis.

5. Please provide detail on the explanatory variables chosen.

6. Please provide citations for the literature review of the explanatory variables.

7. It is unclear why the authors chose to make Medicaid binary. Were participants who did not have insurance included in this population? If so I would encourage the author to recategorize insurance into multiple types (Medicaid, private, none) as this could have an impact of follow up.

8. How were hospital level characteristics chosen as controls? Please provide citations.

9. Please provide more information on the Charlson score. What is included in it and how is it scored? Please provide a citation for it as well.

10. A concern of this paper is how the model is constructed. It is unclear by the aims if the authors are looking for a specific association (ie. Black race) or if they wish to create a general descriptive model (ie. additional patient factors). When describing the statistical methods starting on line 107, the authors seem to create a descriptive model as they include multiple explanatory variables. More clarification and detail is needed on the model construction and goals.

11. Please clarify how the association between early follow up and ED returns was statistically determined. This is missing from the methods section.

12. Please clarify how zipcode-based neighborhood income is constructed.

RESULTS

1. Please clarify in Table 1 what ($1000) means for neighborhood income

2. The author should consider creating a second Table 1 split by Race in the supplementary materials.

3. Please provide a table for the adjusted analysis of early follow up visits.

4. The authors label the variables as “predictors” in Figure 1. However, this is not a predictive model. Please change to “variables”

5. Please make Figure 1 bigger.

DISCUSSION

1. Please provide detail as to what the author means by “although Black patients have worse HF outcomes”.

2. Please provide citations for the sociological variables mentioned on line 186.

3. Please provide strengths of the study.

4. Please provide policy recommendations or implications of the study.

5. Please provide an additional proofread of the manuscript.

Reviewer #2: Overall, I think this paper is shining light on an important issue of inequitable interventions related to heart failure care. My largest critique is that nowhere is the cause of these findings named. There is a rather bland mention of "bias" in the discussion. In my view, what this paper is describing is institutionalized racism, and I believe that should be explicitly stated in this paper. There is plenty of great work to cite and source this discussion from, but perhaps two papers that I might recommend integrating are: (Churchwell et al., 2020; Elias & Paradies, 2021)

Related to this comment, the authors do not justify excluding those outside of the self-reported White and Black race categories. There is one comment that the variables chosen were based on theoretical understanding of risk factors for early emergency department returns, which is conceptually sound. While I am not suggesting that the analysis should have included all self-reported races and ethnicities, I think the authors need to justify this choice, as it may have been found that there is also inequitable care of other racial groups.

Next, the introduction does a better job of differentiating, but in certain places in the paper (such as on line 110) it seems as though the authors include Black race with other markers of clinical vulnerability. Although genetics certainly play a role in heart failure risk for black people (Nayak et al., 2020), what this paper is describing is access to health services rather than genetic risk of HF. If the authors are meaning to talk about genetic risk of HF that should be explicitly stated. I would argue that being Black does not make a person inherently clinically vulnerable. While genetics may predispose a person to some conditions, I do not think that is the phenomenon this paper is focusing on. This needs to be carefully addressed by authors throughout the paper.

Finally, in reading the figures, some interesting protective factors (such as PCP identified in discharge summary or being married) aren't discussed very much- depending on word count, the authors may want to consider adding some text to describe those more in detail.

Minor feedback

• Line 97- three charts? Is this a typo? This seems very low to me but if not please ignore.

• Line 101: Otherwise in the paper heart failure is HF but here it is spelled out, is this intentional?

References

Churchwell, K., Elkind, M. S. V., Benjamin, R. M., Carson, A. P., Chang, E. K., Lawrence, W., Mills, A., Odom, T. M., Rodriguez, C. J., Rodriguez, F., Sanchez, E., Sharrief, A. Z., Sims, M., Williams, O., & On behalf of the American Heart Association. (2020). Call to Action: Structural Racism as a Fundamental Driver of Health Disparities: A Presidential Advisory From the American Heart Association. Circulation, 142(24). https://doi.org/10.1161/CIR.0000000000000936

Elias, A., & Paradies, Y. (2021). The Costs of Institutional Racism and its Ethical Implications for Healthcare. Journal of Bioethical Inquiry, 18(1), 45–58. https://doi.org/10.1007/s11673-020-10073-0

Nayak, A., Hicks, A. J., & Morris, A. A. (2020). Understanding the Complexity of Heart Failure Risk and Treatment in Black Patients. Circulation: Heart Failure, 13(8). https://doi.org/10.1161/CIRCHEARTFAILURE.120.007264

6. PLOS authors have the option to publish the peer review history of their article (what does this mean?). If published, this will include your full peer review and any attached files.

Reviewer #1: **Yes: **Hallie Dau

Reviewer #2: **Yes: **Alexandra Lukey

---

## [Author Response · Author response to Decision Letter 0]

31 Aug 2022

Reviewers' comments:

Reviewer's Responses to Questions

Comments to the Author

1. Is the manuscript technically sound, and do the data support the conclusions?

Reviewer #1: Partly

Reviewer #2: Yes

2. Has the statistical analysis been performed appropriately and rigorously? 

Reviewer #1: I Don't Know

Reviewer #2: Yes

3. Have the authors made all data underlying the findings in their manuscript fully available?

Reviewer #1: No

Reviewer #2: No

4. Is the manuscript presented in an intelligible fashion and written in standard English?

Reviewer #1: Yes

Reviewer #2: Yes

 

5. Review Comments to the Author

Reviewer #1: Summary: Solnick et al. has submitted a study in which they examine if Black race and other patient factors are associated with receiving a scheduled 7-day follow-up after HF hospitalization discharge and whether early follow- up is associated with lower 30-day ED returns. They conducted a retrospective cohort analysis of 6493 Black and White patients at thirteen Michigan hospitals. They reported that early follow-up visits were associated with a lower likelihood of ED returns for HD patients.

INTRODUCTION

1. It is unclear what the author means by “worse outcomes”. What types of outcomes occur in these patients.

This has been clarified: Black patients face the worst heart disease survival rates of all racial groups[1] and in particular, black patients experience higher age-adjusted heart failure (HF) death rates compared to White patients.[

2. Please explain what “transitional care” is.

This is explained now as: “Defined as services a patient receives when changing healthcare settings and including telephone follow-ups or early follow-up visits, transitional care,”

3. It is unclear if there is any literature on early follow up visits and reduced ED times. If so, please cite existing literature. If not, please note that there is no current literature on the matter.

I am unsure of what is meant by “reduced ED times.” Does this mean ED length of stay? If so, there is not literature on this that I am aware of. However, if this is referring to ED visits there is a meta analysis by Vedel et al on the impact of transitional care (varying between telephone calls/ clinic follow-up/ structure home visit ) a 29% overall reduction in ED visits after HR admission. The only one that included clinic follow up was the Duchame paper which did not have a significant effect. The significant studies below were Tsukyiki and Harrison which were a telephone follow-up and structured home visits combined with telephone follow-up (The forest plot for above meta-analysis is copied below for reference). 

4. Please clarify why the authors specifically are examining Black race.

We are examining Black race due to evidence this group has the highest rates of heart failure and evidence there are implicit biases that drive the type of heart failure care they receive. Thus, it may be that they also receive less transitional care and if so, the reason for less transitional care may be bias. The rationale is now in the introduction. 

5. It is not enough to say “additional patient factors” in the objective sentence. Please clarify specific factors that the authors are interested in.

This has been revised as such: “We examine whether Black race, Medicaid, and certain clinical factors were associated with receiving early follow-up after HF hospitalization and whether early follow-up is associated with lower ED returns visits within 30 days.”

The rationale for choosing comorbidities and Medicaid follows similar logic as with selecting Black race, low-income patients and those with comorbidities have higher healthcare utilization for HF, thus if targeted with early follow-ups may have reduced ED return and hospital readmission. 

6. Please note the hypothesis for early follow up and 30-day ED returns

 This has been added to the Intro “We hypothesized that Black and Medicaid patients are less likely to have an early follow-up scheduled, and that early follow-up is associated with lower ED returns”

7. Please provide background information on why it is important to study seven follow ups.

Additional citations including these have been included:

Lee, K. K., Yang, J., Hernandez, A. F., Steimle, A. E., & Go, A. S. (2016). Post-discharge follow-up characteristics associated with 30-day readmission after heart failure hospitalization. Medical care, 54(4), 365.

11,985 HF patients; Used logistic regression. Outpatient contact within 1-7 days after discharge was associated with lower odds of readmission (adjusted odds ratio [OR] 0.81, 95% CI: 0.70–0.94), whereas later outpatient contact between 8 and 30 days after hospital discharge was not significantly associated with readmission (adjusted OR 0.99, 95% CI: 0.82–1.19).

Tung, Y. C., Chang, G. M., Chang, H. Y., & Yu, T. H. (2017). Relationship between early physician follow-up and 30-day readmission after acute myocardial infarction and heart failure. PLoS One, 12(1), e0170061.

The study population for non-ST-segment-elevation myocardial infarction (NSTEMI) and heart failure included 5,008 and 13,577 patients, respectively. Used Cox regression. A completed 7d follow-up was associated with a lower hazard ratio of readmission compared with no early physician follow-up for patients with NSTEMI (hazard ratio [HR], 0.47; 95% confidence interval [CI], 0.39±0.57), and for patients with heart failure (HR, 0.54; 95% CI, 0.48±0.60). They found similar results for a completed 0-14d f/u.

These studies above are cited in this sentence in the introduction “Defined as services a patient receives when changing healthcare settings and including telephone follow-ups or early follow-up visits, transitional care, and specifically 7-day follow-up visits, have been associated with reduced hospital readmissions following HF hospitalization. [16–18]”

8. Please place the study within the context of Michigan and why this study is being conducted in this location.

As for the justification regarding the 13 hospitals:

Hospitals throughout the state of Michigan were recruited, participation was voluntary. Once a hospital agreed to participate, they were given a choice of populations to work with, CHF, COPD or patients transitioning into and out of a SNF.

Context:

The study was conducted in Michigan as Blue Cross Blue Shield of Michigan provides the funding for I-MPACT and other quality collaboratives.

This is described in the Introduction: “We conduct this study in a quality improvement collaborative of 13 hospitals across Michigan which provide data on a set of specific targeted conditions in which HF is one. Hospitals across Michigan were recruited, participation was voluntary. This unique multi-payer database allows for examination of care across a variety of hospital settings with all-payer patient data for HF….. “. Charts were manually abstracted by trained chart abstractors as part of a quality improvement initiative funded by Blue Cross Blue Shield of Michigan (BCBSM) and Blue Care Network

METHODS

1. It is unclear how the authors categorized Black and White. How were mixed race individuals categorized for this study? Furthermore, please clarify how Hispanic individuals were categorized. Finally, please provide justification as to why only Black and White individuals were included in the study.

Only Black and White individuals were included because of prior research identifying the largest racial disparities in HF outcomes between these groups- this is now discussed more thoroughly in the introduction. Patient race was assessed using categories, American Indian or Alaska Native, Asian, Black or African American, Native Hawaiian or Other Pacific Islander, White or Caucasian, and Other Race. For patients who chose, ‘Other Race’ a follow-up question allowed text fields to describe their racial identification, and only one patient identified as mixed race, i.e. ‘Black/African American or White’. The initial sample (N=6,613) was 61.86% White or Caucasian, 36.31% Black or African American while 121 (1.83%) patients belonged to some other race. In the initial sample 45 (0.68%) patients self-identified as Hispanic, 93.35% as Non-Hispanic, and ethnicity data was missing for 395 (5.97%) patients. Due to low number, 121 patients who identified as ‘some other race’ were excluded from analysis. Due to low variation and high missing data in ethnicity, ethnicity could not be considered as a covariate. 

2. Please provide justification as to why the thirteen hospitals were chosen. Additionally, please provide details on these hospitals such as location (urban vs rural), type (public vs private), demographics (race).

Details on urban vs rural was unavailable, but is available for designation of critical access hospital, of which this group includes none. Methods now state: “Details of the involved hospitals are as follows: all non-profit and general acute care hospitals, 6 church ownership, 7 private ownership, 12 teaching hospitals, hospital beds range from 129-637 beds, 0 critical access designated, and distribution of patient race is provided in Figure 1.”

3. Was race the only variable that was self reported?

All data, including race, was abstracted from the medical record. Race was self-reported in the medical record.

4. The author wrote “Early follow-up visits were incentivized by BCBSM through pay-for-performance and were counted if scheduled”. It is unclear if all patients received this incentive or only those that had BCBSM insurance. If the latter, this should be taken into account in the analysis.

Hospitals were incentivized through pay-for-performance to increase scheduling of post-discharge follow-up visits for patients. Patients were not directly incentivized.

5. Please provide detail on the explanatory variables chosen.

Detail on explanatory variable is now included in Supporting Information document under heading “Explanatory Variable Operational Definitions”

6. Please provide citations for the literature review of the explanatory variables.

16. Hernandez AF, Greiner MA, Fonarow GC, Hammill BG, Heidenreich PA, Yancy CW, et al. Relationship between early physician follow-up and 30-day readmission among Medicare beneficiaries hospitalized for heart failure. JAMA. 2010;303: 1716–1722. doi:10.1001/jama.2010.533

17. Lee KK, Yang J, Hernandez AF, Steimle AE, Go AS. Post-discharge Follow-up Characteristics Associated With 30-Day Readmission After Heart Failure Hospitalization. Med Care. 2016;54: 365–372. doi:10.1097/MLR.0000000000000492

18. Tung Y-C, Chang G-M, Chang H-Y, Yu T-H. Relationship between Early Physician Follow-Up and 30-Day Readmission after Acute Myocardial Infarction and Heart Failure. PLoS One. 2017;12: e0170061. doi:10.1371/journal.pone.0170061

21. Hess CN, Shah BR, Peng SA, Thomas L, Roe MT, Peterson ED. Association of early physician follow-up and 30-day readmission after non-ST-segment-elevation myocardial infarction among older patients. Circulation. 2013;128: 1206–1213. doi:10.1161/CIRCULATIONAHA.113.004569

7. It is unclear why the authors chose to make Medicaid binary. Were participants who did not have insurance included in this population? If so I would encourage the author to recategorize insurance into multiple types (Medicaid, private, none) as this could have an impact of follow up.

The main interest was to examine whether patients on Medicaid had lower rates of follow-up compared to patients on Medicare or private insurance. Study sample had 51 (0.79%) patients with ‘No insurance/Self Pay’ and these patients were coded 0 on the binary Medicare measure. 

8. How were hospital level characteristics chosen as controls? Please provide citations.

While we have named the variable on the presence of any transitional care activity a “hospital-level characteristic” (see: Methods section on explanatory variables), we did not include any other hospital-level variables. This is because the IMPACT study did not collect data on hospital-level characteristics. We did not use hospital-level characteristics as controls in models for both study outcomes, post discharge ED visits and scheduling of 7-day follow-up. Of note, we included an indicator variable for each hospital to account for hospital-level heterogeneity, we also used a hospital-level interaction term of Black race and site for an analysis on Black race and early follow-up at a hospital level.

9. Please provide more information on the Charlson score. What is included in it and how is it scored? Please provide a citation for it as well.

Charlson score was calculated according to methodology described in Charson et al (1987). Nineteen conditions used in calculating Charlson score were: Tumor without metastasis (exclude if >5 years from diagnosis), Leukemia (acute or chronic), Lymphoma Metastatic solid tumor, Myocardial infarction (history, not ECG changes only) Congestive heart failure (CHF), Diabetes without end-organ damage (excludes diet-controlled alone), Diabetes with end-organ damage (retinopathy, neuropathy, nephropathy, or brittle diabetes), Mild liver disease (without portal hypertension, includes chronic hepatitis) Moderate or severe liver disease, Cerebrovascular disease (CVA with mild or no residual or TIA) Hemiplegia, Dementia, AIDS (not just HIV positive) Chronic pulmonary disease Connective tissue disease Moderate or severe renal disease Peptic ulcer disease, Peripheral vascular disease (includes aortic aneurysm >= 6 cm). 

Article that developed the Index and one that used it for predicting readmission:

Charlson, M. E., Pompei, P., Ales, K. L., & MacKenzie, C. R. (1987). A new method of classifying prognostic comorbidity in longitudinal studies: development and validation. Journal of chronic diseases, 40(5), 373-383.

Buhr, R. G., Jackson, N. J., Kominski, G. F., Dubinett, S. M., Ong, M. K., & Mangione, C. M. (2019). Comorbidity and thirty-day hospital readmission odds in chronic obstructive pulmonary disease: a comparison of the Charlson and 

These details are provided in: ED Disparities Paper_Statistical Analysis Section.docx

11. Please clarify how the association between early follow up and ED returns was statistically determined. This is missing from the methods section.

The model is clarified by the following addition to methods “Adjusted risk ratios (aRR) were estimated from multivariable logistic regressions evaluating the association between the main outcomes and explanatory variables, with the addition of early follow-up as an explanatory variable in the outcome of ED returns. “

12. Please clarify how zipcode-based neighborhood income is constructed.

These details are provided in Supplement: “Neighborhood income is zip code median household income from the 2006-2010 American Community Survey (https://www.psc.isr.umich.edu/dis/census/Features/tract2zip/), and was measured in units of $1000 for purposes of analysis .

RESULTS

1. Please clarify in Table 1 what ($1000) means for neighborhood income – 

This has been added to the footnotes of the table under caption a: Neighborhood income is zip code median household income measured in units of $1K

2. The author should consider creating a second Table 1 split by Race in the supplementary materials. 

This is now in Supplement Table stratified by race, and added to the results “(and stratified by race in S1 Table)“

3. Please provide a table for the adjusted analysis of early follow up visits. 

This is now added to supplement as S3 table of adjusted analysis of early follow up visit and referenced in results. 

4. The authors label the variables as “predictors” in Figure 1. However, this is not a predictive model. Please change to “variables”– Panel A: predictor changed to Variables; Panel B: predictor changes to Site; and both plots saved as png.

5. Please make Figure 1 bigger.

The terms have been changed to “variables” and “site” PLOS ONE formatting guidelines instructs to safe as TIFF, which we have now done https://journals.plos.org/plosone/s/file?id=wjVg/PLOSOne_formatting_sample_main_body.pdf

DISCUSSION

1. Please provide detail as to what the author means by “although Black patients have worse HF outcomes”.

This is now explained by the first sentence: “Black patients face the worst heart disease survival rates of all racial groups[1] and in particular, black patients experience higher age-adjusted heart failure (HF) death rates compared to White patients.[“ 

2. Please provide citations for the sociological variables mentioned on line 186.

This has been updated to state the variables used in the already cited Roberts [citation 30] study “such as educational attainment, poverty and social isolation.” 

3. Please provide strengths of the study.

This is now addressed in the discussion: “Study strength include: incorporation of detailed patient-level and visit-level variables to help understand contextual factors of healthcare utilization, incorporation of high-quality chart abstracted all-payer data across a wide variety of hospital situated in geographic areas with diverse demographics.”

4. Please provide policy recommendations or implications of the study.

This is now addressed in the discussion as a call to investigate other national quality measures for inequity outcomes and potentially proposed transitional care measures as a process-based alternative, and the importance to measure equity of effect of measures : “ Investigations into transitional care measures that may improve healthcare access is increasingly important given the potential negative effect of influential national policies that may not adequately take into consideration sociodemographic and community factors. [31,32] One such example is with the Centers for Medicare and Medicaid Services (CMS)’s Hospital Readmissions Reduction Program (HRRP), a program which does not incorporate social determinants [33,34] but penalizes hospitals for higher than expected rates of readmission. The policy implications of our findings highlight the need to measure, monitor and mitigate the potential inequitable effects of national quality measures, especially ones that could exacerbate financial challenges for already struggling safety-net hospitals. Incorporating modifiable, process-based measures, such as equity in offering of transitional care services, may represent a future model for federal quality measure consideration. “

5. Please provide an additional proofread of the manuscript.

Have now done so, thank you for your detailed edits and suggestions! 

 

Reviewer #2: Overall, I think this paper is shining light on an important issue of inequitable interventions related to heart failure care. My largest critique is that nowhere is the cause of these findings named. There is a rather bland mention of "bias" in the discussion. In my view, what this paper is describing is institutionalized racism, and I believe that should be explicitly stated in this paper. There is plenty of great work to cite and source this discussion from, but perhaps two papers that I might recommend integrating are: (Churchwell et al., 2020; Elias & Paradies, 2021)

Related to this comment, the authors do not justify excluding those outside of the self-reported White and Black race categories. There is one comment that the variables chosen were based on theoretical understanding of risk factors for early emergency department returns, which is conceptually sound. While I am not suggesting that the analysis should have included all self-reported races and ethnicities, I think the authors need to justify this choice, as it may have been found that there is also inequitable care of other racial groups.

Thank you for these citations. The introduction has been expanded to provide further justification for focusing on black patients- that they have the worst survival rates of all races and there is evidence of a difference in care, and that this difference may be due to bias creates a narrative that there are specific clinical management changes that can be altered to potentially improve the overall health outcomes for black patients. 

Next, the introduction does a better job of differentiating, but in certain places in the paper (such as on line 110) it seems as though the authors include Black race with other markers of clinical vulnerability. Although genetics certainly play a role in heart failure risk for black people (Nayak et al., 2020), what this paper is describing is access to health services rather than genetic risk of HF. If the authors are meaning to talk about genetic risk of HF that should be explicitly stated. I would argue that being Black does not make a person inherently clinically vulnerable. While genetics may predispose a person to some conditions, I do not think that is the phenomenon this paper is focusing on. This needs to be carefully addressed by authors throughout the paper.

We have updated discussion to frame race as marker of social structure impact rather than a marker of genetic predisposition. 

Finally, in reading the figures, some interesting protective factors (such as PCP identified in discharge summary or being married) aren't discussed very much- depending on word count, the authors may want to consider adding some text to describe those more in detail.

We have already added significantly to the word count to discuss race, so unfortunately we will not be able to cover further discussion on this. 

Minor feedback

• Line 97- three charts? Is this a typo? This seems very low to me but if not please ignore.

We have double checked this and not been told otherwise. 

• Line 101: Otherwise in the paper heart failure is HF but here it is spelled out, is this intentional?

Updated to HF.

References

Churchwell, K., Elkind, M. S. V., Benjamin, R. M., Carson, A. P., Chang, E. K., Lawrence, W., Mills, A., Odom, T. M., Rodriguez, C. J., Rodriguez, F., Sanchez, E., Sharrief, A. Z., Sims, M., Williams, O., & On behalf of the American Heart Association. (2020). Call to Action: Structural Racism as a Fundamental Driver of Health Disparities: A Presidential Advisory From the American Heart Association. Circulation, 142(24). https://doi.org/10.1161/CIR.0000000000000936

Elias, A., & Paradies, Y. (2021). The Costs of Institutional Racism and its Ethical Implications for Healthcare. Journal of Bioethical Inquiry, 18(1), 45–58. https://doi.org/10.1007/s11673-020-10073-0

Nayak, A., Hicks, A. J., & Morris, A. A. (2020). Understanding the Complexity of Heart Failure Risk and Treatment in Black Patients. Circulation: Heart Failure, 13(8). https://doi.org/10.1161/CIRCHEARTFAILURE.120.007264

6. PLOS authors have the option to publish the peer review history of their article (what does this mean?). If published, this will include your full peer review and any attached files.

Do you want your identity to be public for this peer review? For information about this choice, including consent withdrawal, please see our Privacy Policy.

Reviewer #1: Yes: Hallie Dau

Reviewer #2: Yes: Alexandra Lukey

---

## [Decision Letter · Decision Letter 1]

13 Oct 2022

PONE-D-21-38345R1Examination of Racial Disparities in Early Follow-up Visits and ED Returns after Heart Failure HospitalizationPLOS ONE

Dear Dr. Solnick,

Thank you for submitting your manuscript to PLOS ONE. After careful consideration, we feel that it has merit but does not fully meet PLOS ONE’s publication criteria as it currently stands. Therefore, we invite you to submit a revised version of the manuscript that addresses the points raised during the review process.

Editor's comments: How did the authors calculate risk ratios when they used the logistic regression analysis?

I agree with the reviewer #1 - the paper is misaligned starting from the title to the selected analytical approach. If the authors are interested in investigating the relationship between race and early follow-up, then a separate model should be built for this purpose (usually, covariates included in the model should be selected based on a solid rationale and literature review – a directed acyclic graph). The second question, if the relationship between Medicaid and early follow-up is investigated, there needs to be another separate model built (again, covariates selection should be based on epidemiological/clinical understanding of causal relationships with other covariates). If the research question is to describe factors associated with early follow-up, then it should be explicitly stated in the research aim and appropriate analysis should be selected without necessary emphasizing that the relationship between race and the outcome is unbiased and the “true” association is derived from this model – as an exploratory models are not designed to investigate each variable independently.

And I could not find the results for the secondary aim investigating the relationship between early follow-up and 30-day ED return.

We look forward to receiving your revised manuscript.

Kind regards,

Alpamys Issanov

Academic Editor

PLOS ONE

Reviewers' comments:

Reviewer's Responses to Questions

**Comments to the Author**

1. If the authors have adequately addressed your comments raised in a previous round of review and you feel that this manuscript is now acceptable for publication, you may indicate that here to bypass the “Comments to the Author” section, enter your conflict of interest statement in the “Confidential to Editor” section, and submit your "Accept" recommendation.

Reviewer #1: (No Response)

Reviewer #2: All comments have been addressed

2. Is the manuscript technically sound, and do the data support the conclusions?

Reviewer #1: No

Reviewer #2: (No Response)

3. Has the statistical analysis been performed appropriately and rigorously? 

Reviewer #1: No

Reviewer #2: (No Response)

4. Have the authors made all data underlying the findings in their manuscript fully available?

Reviewer #1: No

Reviewer #2: (No Response)

5. Is the manuscript presented in an intelligible fashion and written in standard English?

Reviewer #1: Yes

Reviewer #2: (No Response)

6. Review Comments to the Author

Reviewer #1: INTRODUCTION

Please provide an additional proofread of the manuscript. There are some grammatical errors. For example, black is not capitalized on line 86

Lines 117 and 118 should connect to be one paragraph.

This focus of this paper is misaligned. The title of the paper indicates that the focus is on racial disparities. However, the objective seems to focus on Black race and Medicaid as a marker of low-income status. Please clarify and realign.

Low-income status is mentioned in the objective as one of the focuses of the paper. However, it is not discussed in the introduction. Rather the introduction solely focuses on race. Please clarify.

METHODS

Is the missing data range on 143 a typo?

The introduction states that the aim is to examine if Black race and Medicaid are associated with a scheduled 7 day follow up. However, in the methods section numerous explanatory variables are mentioned (demographics, Charlston score, patient hospital characteristics). Many of these seem to be potential confounders rather than explanatory variables (if so, the aim of the study should be changed). Please clarify as this is a major concern.

Please provide an explanation of the sensitivity analysis conducted. The primary analysis uses aRR. However the sensitivity analysis seems to use aOR. Details are needed in the methods section of how this analysis was constructed.

RESULTS

Please review footnote (a) in table 1. It states a linear regression was used for continuous measures. This statistically does not make sense as the data is bivariate. Do the authors possibly mean a t-test?

DISCUSSION

Please provide context of the RCT mentioned on line 294. When and where was it conducted?

Reviewer #2: (No Response)

7. PLOS authors have the option to publish the peer review history of their article (what does this mean?). If published, this will include your full peer review and any attached files.

Reviewer #1: **Yes: **Hallie Dau

Reviewer #2: **Yes: **Alexandra Lukey

---

## [Author Response · Author response to Decision Letter 1]

31 Oct 2022

Editor's comments:

How did the authors calculate risk ratios when they used the logistic regression analysis?

The method used in the manuscript for calculating aRRs following estimation of a logistic model is describe in Norton et al: Norton EC, Miller MM, Kleinman LC. Computing Adjusted Risk Ratios and Risk Differences in Stata. Stata J. 2013;13(3):492-509. doi:10.1177/1536867X1301300304 

I agree with the reviewer #1 - the paper is misaligned starting from the title to the selected analytical approach. If the authors are interested in investigating the relationship between race and early follow-up, then a separate model should be built for this purpose (usually, covariates included in the model should be selected based on a solid rationale and literature review – a directed acyclic graph). The second question, if the relationship between Medicaid and early follow-up is investigated, there needs to be another separate model built (again, covariates selection should be based on epidemiological/clinical understanding of causal relationships with other covariates). If the research question is to describe factors associated with early follow-up, then it should be explicitly stated in the research aim and appropriate analysis should be selected without necessary emphasizing that the relationship between race and the outcome is unbiased and the “true” association is derived from this model – as an exploratory models are not designed to investigate each variable independently.

The purpose of the scheduled follow-up model was not to explore factors associated with this outcome, but to assess the relationship between Black race and scheduled follow-up adjusting for potential confounders. The Statistical Analysis section was revised to make this point clearer. Potential confounders used in model estimation were based on prior literature as cited and were expected to be correlated with race and with the scheduled follow-up. 

However, it should be noted that in the model selection process more emphasis was given to certain confounders. For example, we retained the four LACE risk scoring items in the model regardless of statistical significance, as LACE Instrument is designed for predicting patient risk of mortality and post discharge hospital utilization. ( Based on the work by Van Walraven C, Dhalla IA, Bell C, Etchells E, Stiell IG, Zarnke K, Austin PC, Forster AJ. 2010. Derivation and validation of an index to predict early death or unplanned readmission after discharge from hospital to the community. CMAJ: Canadian Medical Association Journal 182(6):551557 DOI 10.1503/cmaj.091117.) 

And I could not find the results for the secondary aim investigating the relationship between early follow-up and 30-day ED return.

The outcome of ED return has now been highlighted with a new subtitle in the second paragraph of the results section. Logistic model results for relationship between early follow-up and 30-day ED return that were in S1-Fig is now provided in regular Figure 1.

The relationship between early follow-up and 30-day ED return is in fact the first aim of the study, and investigating racial disparities in scheduled follow-up is the second aim. The manuscript is revised to reflect this order. The Statistical Analysis section was revised to explain more clearly the analysis and model estimation. With these revisions we hope to clarify that two logistic models were estimated: (1) a model to evaluate the relationship between ED returns and early follow-up, (2) a model to evaluate the association between Black race and early follow-up (now Fig 2a); and related model stratified by hospital site (now Fig 2b)

6. Review Comments to the Author

Reviewer #1: INTRODUCTION

Please provide an additional proofread of the manuscript. There are some grammatical errors. For example, black is not capitalized on line 86

Thank you for your review, we have conducted an additional proofread of the manuscript, and corrected this error above. 

Lines 117 and 118 should connect to be one paragraph.

This has been corrected. 

This focus of this paper is misaligned. The title of the paper indicates that the focus is on racial disparities. However, the objective seems to focus on Black race and Medicaid as a marker of low-income status. Please clarify and realign.

Racial disparities (between Black and White race) in the offering of early follow-up visits is one of the focuses of the paper, in addition to the association between early follow-up and emergency department returns. Medicaid/ low-income is not the focus of the paper. We will remove Medicaid from the Introduction. 

Low-income status is mentioned in the objective as one of the focuses of the paper. However, it is not discussed in the introduction. Rather the introduction solely focuses on race. Please clarify.

The objective is analyzing the association between early follow-up and emergency department returns, and of race and early follow up. Low-income status has been removed from the objective. 

METHODS

Is the missing data range on 143 a typo?

Changed text to: Missing data ranged from 0% to 1.89%

The introduction states that the aim is to examine if Black race and Medicaid are associated with a scheduled 7 day follow up. However, in the methods section numerous explanatory variables are mentioned (demographics, Charlston score, patient hospital characteristics). Many of these seem to be potential confounders rather than explanatory variables (if so, the aim of the study should be changed). Please clarify as this is a major concern.

Medicaid is considered as a confounder, not an explanatory variable, and the manuscript has been changed accordingly. As described above, apart from the variables of interest (which are described in the revised Statistical Analysis section) the remaining factors including four LACE risk scoring items, prior hospital admissions, other clinical and demographic variables were considered as confounders and not as explanatory variables. For the outcome ED returns the variable of interest (explanatory variable) was scheduled follow-up. For the outcome scheduled follow-up the variable of interest was race.

Please provide an explanation of the sensitivity analysis conducted. The primary analysis uses aRR. However the sensitivity analysis seems to use aOR. Details are needed in the methods section of how this analysis was constructed.

Sensitivity analysis was conducted to see whether findings were sensitive to exclusion of patients discharged during the Covid pandemic. Sensitivity analysis was conducted by re-fitting the main regression models using a sample that excluded patients discharged from April 1, 2020 through the end of sample data on September 30, 2020. 

For consistency, the sensitivity analysis is now in aRR not aOR in the paragraph before ‘Discussion’.

RESULTS

Please review footnote (a) in table 1. It states a linear regression was used for continuous measures. This statistically does not make sense as the data is bivariate. Do the authors possibly mean a t-test?

 T-tests were completed for continuous measures and Table 1 footnote text was revised to “… p-values for continuous measures are based on T-tests”.

DISCUSSION

Please provide context of the RCT mentioned on line 294. When and where was it conducted?

This is now described in location and with more detail: “. Our theory—that transitional care can avert ED returns—is supported by previous randomized trials based in Canada, which found that HF patients randomized to transitional care after discharge had a reduction in rate of ED returns [19]: in one study in 2000 by Harrison et al. of 200 patients, nurse-led transitional care of structured home visits and telephone follow-up reduced ED visits; in the second study in 1998 by Tsuyuki et al. of 276 patients, transitional care including patient education and telephone follow-up reduced cardiovascular-related ED visits.”

---

## [Editor Report · Decision Letter 2]

16 Nov 2022

PONE-D-21-38345R2Emergency department returns and early follow-up visits after heart failure hospitalization: cohort study examining the role of racePLOS ONE

Dear Dr. Solnick,

Thank you for submitting your manuscript to PLOS ONE. After careful consideration, we feel that it has merit but does not fully meet PLOS ONE’s publication criteria as it currently stands. Therefore, we invite you to submit a revised version of the manuscript that addresses the points raised during the review process.

We look forward to receiving your revised manuscript.

Kind regards,

Alpamys Issanov

Academic Editor

PLOS ONE

Journal Requirements:

Additional Editor Comments:

I appreciate the re-analysis conducted by the authors. There are a few things that need clarification:

In the statistical analysis, the authors stated that logistic regression models were built to answer the research questions, however, all estimates were presented as adjusted relative ratios, not odds ratios. Could the authors provide the explanation how aRR were calculated from logistic regression models?

It looks like the authors also had one additional research objective: "To test the hypothesis that the association between Black race and early follow-up may vary among across sites..." Please show this research aim in the introduction as well and present the corresponding findings in the results.
---

## [Author Response · Author response to Decision Letter 2]

23 Nov 2022

Question: 

In the statistical analysis, the authors stated that logistic regression models were built to answer the research questions, however, all estimates were presented as adjusted relative ratios, not odds ratios. Could the authors provide the explanation how aRR were calculated from logistic regression models?

Response: 

We have re-written the methods section to clarify that the ARR are calculated using the results of the logistic regression, reconfigured as a relative risk to be easier to interpret. This is a method that is widely used in medical literature as papers have cited this original article describing the methodology and STATA commands : Norton et al: Norton EC, Miller MM, Kleinman LC. Computing Adjusted Risk Ratios and Risk Differences in Stata. Stata J. 2013;13(3):492-509. doi:10.1177/1536867X1301300304 

Per this paper, Norton et al describes: [copied from the Norton manuscript here for convenience: “The ARR and ARD are two ways to express the relationship between two predicted probabilities based on the fit model and a set of observations. One is the predicted probability when the variable of interest equals 1; the other is the predicted probability when the variable of interest equals 0 (more generally, pick any two values of the variable). These predicted probabilities are then averaged over the entire dataset (or perhaps an interesting subset of the data). The ARR is the ratio of the mean predicted probabilities.” 

Question:

It looks like the authors also had one additional research objective: "To test the hypothesis that the association between Black race and early follow-up may vary among across sites..." Please show this research aim in the introduction as well and present the corresponding findings in the results.

Response:

We have now added this research aim of analyzing Black race and early follow-up by hospital site into the introduction. (Added: “Because transitional care practices may vary by hospital, we further examine whether Black race is associated with early follow-up rates at the hospital-level.”) 

 It is also now included in the hypothesis (“We hypothesized that early follow-up is associated with lower ED returns and that Blacks are less likely to have an early follow-up scheduled overall, but that there would be lower rates of follow-up at hospitals with less Black patients. “) 

As for the methods, we have drawn distinction by creating a new paragraph and more details in describing this part. For further methodological details: The interaction model to calculate the site-specific estimates is done as follows: The model has a binary indicator for each of the 13 hospital sites and an interaction term between each site and Black race. The fitted model with the interaction term is used to calculate the predicted probability for follow up for Black patients specific to each site. 

As for the results, we clarified which paragraph is discussing the early follow-up by hospital site, and re-wrote the introduction to that section to be clearer.

---

## [Editor Report · Decision Letter 3]

7 Dec 2022

Emergency department returns and early follow-up visits after heart failure hospitalization: cohort study examining the role of race

PONE-D-21-38345R3

Dear Dr. Solnick,

We’re pleased to inform you that your manuscript has been judged scientifically suitable for publication and will be formally accepted for publication once it meets all outstanding technical requirements.

Kind regards,

Alpamys Issanov

Academic Editor

PLOS ONE
---

## [Editor Report · Acceptance letter]

14 Dec 2022

PONE-D-21-38345R3 

Emergency department returns and early follow-up visits after heart failure hospitalization: cohort study examining the role of race 

Dear Dr. Solnick:

I'm pleased to inform you that your manuscript has been deemed suitable for publication in PLOS ONE. Congratulations! Your manuscript is now with our production department. 

Kind regards, 

on behalf of

Dr. Alpamys Issanov 

Academic Editor

PLOS ONE